# An Explanation of Exceptions from Chargaff’s Second Parity Rule/Strand Symmetry of DNA Molecules

**DOI:** 10.3390/genes13111929

**Published:** 2022-10-23

**Authors:** Marija Rosandić, Ines Vlahović, Ivan Pilaš, Matko Glunčić, Vladimir Paar

**Affiliations:** 1University Hospital Centre Zagreb (Ret.), 10000 Zagreb, Croatia; 2Croatian Academy of Sciences and Arts, 10000 Zagreb, Croatia; 3Faculty of Science, Algebra University College, 10000 Zagreb, Croatia; 4Forest Research Institute, 10450 Jastrebarsko, Croatia; 5Physics Department, Faculty of Science, University of Zagreb, 10000 Zagreb, Croatia

**Keywords:** coding DNA, noncoding DNA, DNA quadruplets, DNA symmetries, trinucleotide classification, human chromosomes, Neuroblastoma BreakPoint Family genes

## Abstract

In this article, we show that mono/oligonucleotide quadruplets, as basic structures of DNA, along with our classification of trinucleotides, disclose an organization of genomes based on purine–pyrimidine symmetry. Moreover, the structure and stability of DNA are influenced by the Watson–Crick pairing and the natural law of DNA creation and conservation, according to which the same mono- or oligonucleotide insertion must be inserted simultaneously into both strands of DNA. Taken together, they lead to quadruplets with central mirror symmetry and bidirectional DNA strand orientation and are incorporated into Chargaff’s second parity rule (CSPR). Performing our quadruplet frequency analysis of all human chromosomes and of Neuroblastoma BreakPoint Family (NBPF) genes, which code Olduvai protein domains in the human genome, we show that the coding part of DNA violates CSPR. This may shed new light and give rise to a novel hypothesis on DNA creation and its evolution. In this framework, the logarithmic relationship between oligonucleotide order and minimal DNA sequence length, to establish the validity of CSPR, automatically follows from the quadruplet structure of the genomic sequence. The problem of the violation of CSPR in rare symbionts is discussed.

## 1. Introduction

In 1951, Chargaff´s first parity rule on nucleotide pairing in DNA molecules was published [1]. This statement on the equality of frequencies of nucleotides A and T and of C and G in the whole DNA molecule was a guideline for the discovery of the structure of DNA. Watson and Crick (1953) [2] explained this pairing as a double-helix structure, where the two chains of DNA are connected by hydrogen bonds of A with T and C with G. In 1968, the unexpected Chargaff´s second parity rule (CSPR) was published as an empirical global rule for long enough segments of chromosomes. CSPR states a marked similarity of frequencies of nucleotides A and T and of C and G, even within each of the two strands of DNA [3]. It is noted that, for such a rule, not being derived from a compelling principle such as base pairing, underlying the first rule remains a challenge. This rule was extended to the similarity of frequencies of oligonucleotides to those of their respective reverse complements within each DNA strand in segments that are long enough (>100 kb for trinucleotides, the best studied oligonucleotides) [4,5,6,7,8,9,10,11,12,13,14,15,16,17,18,19,20,21]. In the literature, various other names have also been used for CSPR, such as “strand symmetry”, “intra-strand symmetry”, “word symmetry” and “inversion symmetry”. According to its meaning, this rule can also be called Chargaff nonlocal pairing. For 50 years, a conclusive explanation of CSPR has still been rather controversial [7,8,11,12,14,15,16,17,18,21,22,23,24,25,26,27,28,29,30,31,32,33,34,35].

CSPR implies that direct mononucleotide/oligonucleotide–reverse complement mononucleotide/oligonucleotide equality is valid up to a statistical limit set by the length of the examined sequence. CSPR for oligonucleotides cannot be explained solely by mononucleotides or lower-order oligonucleotide CSPR. Therefore, it has been suggested that it is likely that sequences were initially written at the oligonucleotide level under the evolutionary forces that required a parity at that level [19,28,35]. For the exact validity of CSPR, genes on both DNA strands must be exactly equal, and each sequence on the top strand (read 5′→3′) must correspond to the same gene sequence on the bottom strand (read 3′→5′). With deviations from CSPR, the difference between the corresponding genes on the top and bottom strand increases (Shporer at al. 2016). The empirically found Szybalski’s rule states that, in bacteriophage coding sequences, purines (A and G) exceed pyrimidines (C and D), which violates CSPR in the coding region [36]. Szybalski’s rule has also been confirmed in other organisms, violating CSPR in favour of purines [28].

The symmetry associated with CSPR is related to a broader framework of symmetries in science. The general question is whether symmetries reflect some fundamental “laws” of genome evolution or whether they are a type of statistical pattern [37]. The idea that natural laws are associated with some symmetry is widespread, but the symbiosis of mathematics and natural laws is not yet fully understood [38,39].

In a broader framework, a spectacular result was achieved by Emmy Nöther in 1918, proving her famous theorem by relating symmetry in time to the energy conservation law [40,41]. Gross expressed a general remark on the symmetry principle as a feature of nature: “We are embarked on a new stage of exploration of fundamental laws of nature, a voyage guided largely by the search for discovery of new symmetries” [42]. The symmetry concept also spread in biology, including in genomics [17,19,31,38,39,40,41,42,43,44,45,46,47,48,49,50,51,52]. Jacques Monod attached great significance to symmetry in biology. He pointed out that symmetry must not be understood in purely geometrical connotations, but rather in a much wider sense: “The concept of symmetry becomes almost identical with that of order within a structure, whether in space or time, or purely *in abstracto*. The difficulties stem precisely from the extreme complexity of biological order, even though it often does express itself, partially, in some very simple and very obvious symmetry elements.” [43].

We propose an explanation of the work of Rosandić et co-workers [19]. Each of the possible 20 trinucleotide quadruplets consist of direct (D), reverse complement (RC), complement (C) and reverse (R) trinucleotides. Due to mirror symmetry, our classification of trinucleotides (Table 1) has an embedded CSPR [17,18,19,50,51]. It should be stressed that, regardless of how many times a quadruplet is multiplied, CSPR is not violated and remains integrated into the DNA genome.

We prove the persistence of quadruplet mirror symmetries empirically, analyzing the whole genome in prokaryotes for free-living bacteria, in archaea and in the chromosomes of some eukaryotes, from one of the smallest *Sacharomyces cerevisiae* all the way to the modern *Homo sapiens sapiens* and extinct *Homo sapiens neanderthaliensis* [18,19]. In our opinion, in such a long evolutionary period, the strictly controlled DNA structure with quadruplet symmetries ensuing CSPR was preserved due to the *natural symmetry law of DNA creation and conservation* [18,19,50,51]. Accordingly, the same mutation (insertion) for the mononucleotide or oligonucleotide as a random event must encompass both strands of DNA, regardless of localization (Figure 1). In this way, the integrity of quadruplet symmetries persists within DNA, i.e., the simultaneous identical growth of both strands of DNA is enabled.

However, four possible exceptions deserve attention. First, CSPR is not fulfilled for trinucleotide sequences shorter than about 100 kb. By further decreasing the sequence length to about 50 kb, the difference between frequencies *f*(D) and *f*(RC(D)) increases, and for smaller lengths, any tendency of *f*(D) and *f*(RC(D)) frequency identity disappears. Second, CSPR gradually disappears with an increase in the number of quadruplets’ oligonucleotides. In each human chromosome, the frequencies *f*(D) and *f*(RC(D)) for trinucleotides differ by less than 1%. For higher order oligonucleotides with up to six constituting nucleotides, this difference gradually increases, and with ten nucleotides, the frequencies *f*(D) and *f*(RC(D)) differ sizably from each other, i.e., CSPR does not hold any more.

Third, our study shows that, for the coding DNA of any human chromosome, CSPR is not satisfied. Since the DNA of the whole human chromosome is approximately characterized by CSPR (at the level of deviation below 1%), it appears that the largest part of this deviation from CSPR arises from the coding DNA, which is less than 2% in the human genome. Fourth, genomes of some rare prokaryotes deviate from CSPR: *Candidatus tremblaya princeps*, *Candidatus hodgkinia cicadicola*, *Filifactor alocis* and *Pseudovibrio_sp.FO-BEG1*, all of them being symbionts, as is discussed later. In this study, we aim to explain the origin of these CSPR exceptions. It is necessary to explain our results in investigating quadruplet symmetries and quadruplet frequency analyses of the DNA molecule.

## 2. Materials and Methods

The data (DNA sequences) that were used for this analysis are:(1)*Candidatus_Carsonella_ruddii, GCF_000287255.1_ASM28725v1,* (ftp://ftp.ncbi.nlm.nih.gov/genomes/refseq/bacteria/Candidatus_Carsonella_ruddii/latest_assembly_versions/GCF_000287255.1_ASM28725v1, accessed on 18 September 2022).(2)Human DNA sequence- RefSeq assembly accession: GCF_000001405.33., GRCh38.p7.(3)NBPF family gene.(4)*C. tremblaya princeps* (annotation GCF_000219195.1_ASM21919v1, ftp://ftp.ncbi.nih.gov/genomes/refseq/bacteria/Candidatus_Tremblaya_princeps/all_assembly_versions/suppressed/, accessed on 18 September 2022).(5)*F. alocis (annotation GCA_000163895.2 ASM16389v2, https://www.ncbi.nlm.nih.gov/nuccore/NC_016630.1s,* accessed on 18 September 2022).(6)Pseudovibrio_sp.FO-BEG1 (GCA_000236645.1_ASM23664v1, ftp://ftp.ncbi.nlm.nih.gov/genomes/genbank/bacteria/Pseudovibrio_sp._FO-BEG1/latest_assembly_versions/GCA_000236645.1_ASM23664v1, accessed on 18 September 2022).

For calculating trinucleotides, we use a custom-made computational program in C# that calculates trinucleotide frequencies from DNA sequences for nucleotides by using a sliding window (we neglect trinucleotides containing *N* bases). The computational method CLT__Find, which is used in the calculations, is available at the following link: http://genom.hazu.hr/tools.html, accessed on 18 September 2022.

We use a custom-made program in python for random sequences, where we use sequences from real DNA (from species mentioned in this paper). With this program, we select 1000 sequences/subsets picked from DNA without overlapping between them from a random start position in each human chromosome. Each subset sequence is 200 bp in length. Sequences obtained in this way are concatenated, and the CLT_Find algorithm is applied for their analysis (http://genom.hazu.hr/tools.html, accessed on 18 September 2022).

In regression analysis, the term “standard error” refers either to the square root of the reduced chi-squared statistic or the standard error for a particular regression coefficient (e.g., as used in confidence intervals). The standard error bounds are computed using the predicted locally estimated scatterplot smoothing (LOESS) method. The exception is LOESS, which uses a t-based approximation. The gray area around the line of equalizing presents confidence interval of the estimate.

We show that the CSPR analysis of triplets in hg38 human genomic sequences preserves CSPR as a global pattern in the noncoding part of DNA, and it violates CSPR in the coding part. We use the hg38 assembly because, when we started with this study, the available assembly sequence was incomplete. Only recently, the complete reference assembly T2T-CHM13 opened new opportunities to investigate the role of genome organization and regulation [53,54]. We argue that the extension of the analysis to the T2T-CHM13 assembly cannot significantly influence our obtained results by using the hg38 genomic assembly. For coding parts of DNA, this is obvious, since genes are nearly gap free and already well sequenced in hg38, giving results like to the T2T-CHM13 assembly. For example, we check this for NBPF genes with relatively long NBPF exons in human chromosome 1. In sizable sequenced noncoding segments, no CSPR violations are found, both in continuous arrays and in concatenated widely scattered randomized segments. In many of such cases, these structures are characterized by repeated patterns and their deviations of similar types in hg38 sequenced sections and in those which are sequenced only by T2T-CHM13. Such comparisons have been considered, for example, for human chromosomes 21 and Y.

## 3. Results and Discussion

### 3.1. Distinction between DNA “Strand Symmetry” and “Quadruplet Symmetries”

A distinction should be made between treating DNA strand symmetry and DNA quadruplet symmetry. Strand symmetry for individual nucleotides refers to the equality of A to T frequencies and C to G frequencies within each DNA strand (Figure 1C). Analogously, the strand symmetry for trinucleotides (Figure 2A) refers to the equality of any trinucleotide (denoted D) to its reverse complement (denoted RC(D)) within each DNA strand. From 64 possible trinucleotides, there is one group of 32 trinucleotides that are Ds, and the remaining group of 32 trinucleotides comprises their respective reverse complements RC(D)’s. If the frequency of each trinucleotide D from the first group is approximately equal to the frequency of its RC(D) from the second group, then CSPR is valid for trinucleotides. In this way, DNA is reduced to a binary system. On the other hand, looking at this bidirectionally, the same combination of D and RC(D) appears in both strands (Figure 2A). Therefore, usually only one strand of DNA is analysed, and the term “strand symmetry” is used as a synonym for CSPR. Quadruplet symmetry with unidirectional reading of both DNA strands represents a quartic system, which is for 64 trinucleotides structured in 20 specific quadruplets according to our trinucleotide classification (Table 1) [17,19,50,51]. As has already been indicated, each quadruplet consists of D, RC(D), C(D) and R(D) of a mononucleotide or of an oligonucleotide denoted as D, e.g., a trinucleotide (Figure 2B). The role of a direct mononucleotide or oligonucleotide can have any member of the quadruplet. For any D, the corresponding RC(D), C(D) and R(D) members are formed, but the same quadruplet always consists of the same members, e.g., trinucleotides. Moreover, trinucleotides of the same quadruplet are either A+T rich or C+G rich (Table 1).

In most of our analyses, we use trinucleotide frequencies because, in coding DNA, there are 60 possible codons and 4 signals: the start signal (AUG), which specifies the amino acid methionine but serves in certain contexts as the initiation codon, and 3 stop signals (UGA, UAG and UAA). Dinucleotides, with 16 possible combinations and only 6 different quadruplets, are restricted in information content. Mononucleotides, which are even more restricted, have only 2 quadruplets: the one composed of A and T, and the other of C and G nucleotides.

On the other hand, oligonucleotides composed of 4 nucleotides and 256 possible combinations give 68 possible quadruplets (Table 2), which would unnecessarily complicate the analysis. In this sense, it appears that the genome is being gauged for trinucleotides. Therefore, in this analysis, we use trinucleotides and mononucleotides.

Each quadruplet consists of structural purine/pyrimidine symmetries of the mirror type within each strand and between both strands of DNA (Figure 2B,C). It should be stressed that each A+T-rich quadruplet (Table 1) consisting of trinucleotides is composed of three different nucleotides, but they must contain A and T (for example, D = ATG, RC(D) = CAT, C(D) = TAC and R(D) = GTA) as well as 4A, 4T, 2C and 2G nucleotides in each strand. Each C+G-rich quadruplet (Table 1) (for example, D = CGT, RC(D) = ACG, C(D) = GCA and R(D) = TGC) contains, according to the same scheme, 4C, 4G, 2A and 2T nucleotides in each strand. The quadruplets consisting of only A and T nucleotides (for example, D = TAA, RC(D) = TTA, C(D) = ATT and R(D) = AAT), of only A nucleotides or of only T nucleotides (for example, D = AAA, RC (D) = TTT, C(D) = TTT and R(D) = AAA) contain 6A and 6T nucleotides in each strand. Analogously, the quadruplets consisting of only C and G nucleotides (for example, D = GCC, RC(D) = GGC, C(D) = CGG and R(D) = CCG) of only C nucleotides or of only G nucleotides (for example, D = CCC, RC(D) = GGG, C(D) = GGG and R(D) = CCC) contain 6C and 6G nucleotides in each strand of DNA. This reflects the internal symmetry of our classification of trinucleotides (Table 1).

Quadruplets, as basic structural symmetry elements of DNA molecules, automatically ensure the CSPR pattern [17,19,50,51], regardless of how many times they have been individually multiplied. Accordingly, the relative frequency of every member of the quadruplet is almost equal within both strands of DNA (Figure 2D). All members of a quadruplet within the same quadruplet box (Q box) have almost the same relative frequency, but it differs between both Q boxes (Qbox_D-RC_ and Qbox_C-R_), except for symmetric trinucleotides, where it is equal (for example, D = ACA, RC(D) = TGT, C(D) = TGT and R(D) = ACA).

The quadruplet symmetry structure shows that the same trinucleotides appear in Qbox_D-RC_ and in Q-box_C-R_, but in the opposite strands (Figure 2B). On the other hand, “strand symmetry” does not recognize their common structure, and the members of each Q box are treated as independent (Figure 2A). The “strand symmetry” analysis is focused on only one strand of DNA and does not distinguish the quadruplet structure by its purine/pyrimidine mirror symmetries between both strands and within each strand of the same quadruplet, and neither does it realize the same frequency in both strands of each of the four members of the same quadruplet (Figure 2D).

For each prokaryote genome and each eukaryote chromosome, we determine the corresponding A+T-rich matrices with 10 A+T-rich quadruplets and C+G-rich matrices with 10 C+G-rich quadruplets. We find an inverse proportional relation between relative frequencies in Qbox_D-RC_ and Qbox_C-R_ within the same quadruplet and an inverse proportional relation between A+T-rich and C+G-rich matrices through their purine–pyrimidine relations. This can be also recognized in histograms included as illustrations in the text (Figure 3, Appendix A). The inverse proportionality relationship of the frequency between trinucleotides within each quadruplet and between A+T-rich and C+G-rich quadruplet matrices preserves the integrity of the quadruplet and of the whole genome. Randomly taking four trinucleotides (or some other oligonucleotides, or mononucleotides) cannot create quadruplets with mirror symmetries and cannot satisfy CSPR despite being in accordance with the Watson–Crick parity rule. We also call the quadruplet symmetry “butterfly symmetry” because of its highly visible aesthetics.

### 3.2. An Approximate Rule for the Minimal Sequence Length for the Determination of CSPR Symmetry

It is known empirically that, using trinucleotides, CSPR is approximately satisfied with differences between the frequency of direct and reverse complement trinucleotides below 2% for sequence lengths of at least 100 kb [8,10,16,45,55]. Based on this empirical fact, it follows that each trinucleotide must appear in this sequence ~1500 times on average (100,000 bp:64 = 1562.5). We take this as an empirical gauge for a minimal sequence length required to determine CSPR symmetry. Using this gauge for oligonucleotides containing *n* nucleotides, there are 4*^n^* combinations for the corresponding quadruplet members, i.e., 4^1^ = 4 for mononucleotides, 4^2^ =16 for dinucleotides, 4^3^ = 64 for trinucleotides, 4^4^ =256 for tetranucleotides, 4^10^ = 1,048,576 for decanucleotides (*n* =10), (second and third column in Table 2). Accordingly, with increases in the number of nucleotides in oligonucleotides, the number of different oligonucleotides of the same order increases rapidly. For every increase in oligonucleotide order by one, the number of *n* nucleotide combinations increases by four, reflecting the quartic pattern D-RC(D)-C(D)-R(D) of each quadruplet.

How does it influence the minimal length *L_n_* of genomic sequences required for the approximate persistence of CSPR symmetry using an oligonucleotide of *n* nucleotides? As a simple guideline, we can assume that the increase in oligonucleotide order by one should be accompanied by an increase in the minimal sequence length by four, which is the number of members in a quadruplet. Adopting the empirically based gauge for the trinucleotide case (*n* = 3), we can obtain the minimal sequence lengths given in Table 2 (third column). We note that using simplicity as a problem-solving principle (generally referred to as Occam´s razor) is frequently used as a guideline in the history of science [56] (Gauche, 2003). Here, it gives a rapid increase in the minimal sequence length required for CSPR investigations. Thus, for example, for decanucleotide oligonucleotides (*n* =10), the minimal sequence length for the determination of CSPR should be about 1600 Mb (Table 2).

Thus, we can obtain an approximate formula for the dependence of the minimal sequence length (*L_n_* ≈ 1500 … 4*^n^*^−3^). It follows that the order of the minimal trinucleotide *n* depends logarithmically on the length of the minimal length of the required genomic sequence (*n ~* log *L_n_*) (fourth column in Table 2, Figure 4). The logarithmic dependence of the minimal oligonucleotide order on the length of the required genomic sequence is suggested based on previous computations [57]. We note that such a simplified assumption may correspond to the lowest limit of the length estimate, and more elaborate considerations can lead to some enlargements of size.

Previous studies of human DNA sequences have shown that, in accordance with CSPR, the frequencies of direct and reverse complement oligonucleotides are approximately equal up to oligonucleotides with 6 nucleotides. With increases to higher order oligonucleotides to 7, 8 and 9 nucleotides, the differences between the frequencies of D and RC(D) become bigger, and after oligonucleotides with 10 nucleotides, the frequency differences change radically and sporadically and gradually break down, instead of breaking abruptly [35]. As seen, the answer to the question of why oligonucleotides with 10 or more nucleotides do not satisfy CSPR is shown in Table 2. Namely, with increases in the number of nucleotides, the number of possible combinations among oligonucleotides (as well as the number of different quadruplets) increases, and a longer DNA sequence is needed that can satisfy CSPR. For a direct oligonucleotide of 6 nucleotides and its 4096 possible combinations, the sequence of 6.4 million nucleotides is needed in this approach (Table 2), e.g., the genome size of many bacteria is within 6 million bp. As we have shown, each combination of trinucleotides must appear at least approximately 1500 times to confirm CSPR.

In investigations of the human genome, each chromosome is usually investigated individually. Therefore, CSPR is determined for an individual chromosome if there is a direct oligonucleotide in a quadruplet, even with 8 nucleotides. For an estimate of 65,536 possible combinations of oligonucleotides, an analysis of the DNA sequence of about 102 million nucleotides is needed, and this order of magnitude corresponds to most human chromosomes. The problem arises for oligonucleotides of 9 nucleotides, because in the corresponding minimal sequence, a length of about 4 thousand million bp is needed, whereas the largest human chromosome has only about 250 million. For 10-nucleotide oligonucleotides with 1,048,576 possible combinations, a sequence of over 1600 million nucleotides is needed to determine whether it satisfies CSPR. This means that half of the whole human genome is needed. For 11-nucleotide oligonucleotides, the CSPR analysis requires more than 6500 million nucleotides, i.e., the whole human genome would be too short. Such a gradual increase to longer DNA sequences for higher order oligonucleotides, because of the rapid growth of the number of oligonucleotide combinations, provides an explanation of why “the strand symmetry would break up gradually instead of breaking abruptly”, showing that strand symmetry persists for oligonucleotides of up to 9 nucleotides in the human genome for its oligonucleotide frequency pattern [35,55].

On the other hand, in sequences smaller than 100 kb, there are not enough individual trinucleotides to ensure the equality of frequencies *f*D = *f*RC(D) at a level of deviation below 2%. Such sequences have a quadruplet structure, but at first glance, they do not satisfy CSPR. In smaller sequences below 100 kb, it is more appropriate to investigate whether CSPR is satisfied for dinucleotides, which, with their 16 possible combinations, have only six quadruplets. For example, one of them is D = AC, RC (D) = GT, C(D) = TG and R(D) = CA. To determine whether CSPR is satisfied for dinucleotides, we can already empirically find that the 25 kb sequence is long enough (Table 2).

For mononucleotides (*n* = 1) the CSPR is satisfied if A and T in the sequence have the same frequency of *f*(A) = *f*(T), and similarly, of C and G, *f*(C) = *f*(G) (at a level of deviation below 2%). Starting with our gauge in the case of trinucleotides, for mononucleotide sequences of about 6 kb, they already have sufficient length to be in accordance with CSPR (Table 2).

### 3.3. Breakdown of CSPR and Quadruplet Symmetries in Coding DNA

In studies of trinucleotides in DNA sequences of >100 kb, it is seen that the frequency of each trinucleotide D is approximately equal to the frequency of its reverse complement RC(D) within the same strand (at a level of deviation below 2%). Here, we investigate whether this deviation appears as a statistical error or has some other explanation. To answer this question, with our quadruplet frequency method for trinucleotides, we analyze all relative frequencies of A+T-rich and C+G-rich quadruplet matrices in each human chromosome (Appendix A). We obtain *f*(D) = *f*(RC) within each strand at a level of deviation about 0.5%. Simultaneously, we study differences in relative percentages between complementary A-T and C-G base pairs of all human chromosomes. This difference does not exceed 0.33%. This agreement with CSPR illustrates Figure 3A and Appendix A. There is a high degree of similarity in relative frequencies of complementary pairs A-T or C-G bases in chromosomes 1–12, X and Y, and these chromosomes have submetacentric or acrocentric shapes rather than regular metacentric shapes. Except for the Y chromosome, smaller and more irregular telocentric chromosomes 13–22 mutually differ more in quadruplet relative frequencies, but they approximately satisfy CSPR (Appendix A).

For the control group, we randomly select from each chromosome group of 200 nucleotides, forming a combined random sequence of 200,000 bp. In this way, we investigate whether, for each chromosome, a random sequence constructed of 1000 randomly selected, scattered and small DNA segments of 200 nucleotides concatenated into a test sequence, retains the CSPR pattern as the whole chromosome does. We show that, indeed, the frequency analysis of each random concatenated sequence approximately satisfies CSPR (Figure 3B, Appendix A). With our quadruplet frequency method for trinucleotides, we show that the frequency matrices from all whole human chromosomes and their random concatenated sequences are mutually very similar (Appendix A).

However, the CSPR situation is different for coding DNA; a combined sequence of all exons concatenated into a whole coding sequence, in each human chromosome, does not satisfy CSPR (Figure 3C), although CSPR is satisfied for whole DNA sequences in each chromosome individually (Appendix A). This sheds a new light on the pattern of DNA. It should be noted that such combined coding DNA of each chromosome contains from 350,406 bp (Y chromosome) to 17,202,568 bp (chromosome 1), which far exceeds the length of 100,000 bp needed for a reliable analysis (Appendix A). Thus, the concatenated coding DNA of each human chromosome does not satisfy CSPR; there are substantial differences in relative frequencies between A and T and between C and G nucleotides.

We show the quadruplet frequency analysis of A-T-rich and C-G-rich quadruplet matrices of trinucleotides in human chromosome 1 (Figure 3, Appendix A) and the differences between sequences, which are not in accordance with CSPR. For an analysis of a 248,956,422 bp sequence of human chromosome 1 (Figure 5A), which is in accordance with CSPR, a 200,000 bp sequence of a randomly selected subsequence within the same chromosome (Figure 5B) is concatenated, which is also in accordance with CSPR, and within the same coding DNA (concatenated 17,202,568 bp of the subsequence within chromosome 1) (Figure 5C, Appendix A), which is not in accordance with CSPR. The analyses of all human chromosomes with the same results are presented in Appendix A.

In a further step, we also analysed whole genes and their exons and introns for all Neuroblastoma BreakPoint Family (NBPF) genes, which code Olduvai protein domains [58] in the human genome (Appendix A). This group of genes contains 16 genes in the interval of 10,845 bp–166,939 bp and 6 pseudogenes in the interval of 4543 bp–41,626 bp. Our analysis of the relative sequences of A and T and of C and G nucleotides shows that, in the same strand of DNA, CSPR is not satisfied for individual genes (Figure 5D, Appendix A), for their introns (Figure 5E, Appendix A) and for their exons (Figure 5F, Appendix A). This is also clearly seen with our quadruplet analysis on the histograms of trinucleotide quadruplet matrices of NBPF genes for the 19th gene of 166,939 bp, 14th gene of 149,567 bp, 26th gene of 118,759 bp and 20th gene of 117,079 bp, which are long enough (>100,000 bp) to give significant results (Appendix A). With the same analysis, CSPR is violated in the long-enough introns of 19, 20, 21P and 26 genes (Appendix A) in NBPF genes, which code Olduvai protein domains in the human genome [58]. As we show, the analysis of intron mononucleotides for NBPF genes also has a violation of CSPR; within the same strand, the frequencies of A and T complements as well as of C and G complements sizably differ (Figure 5E). Unfortunately, exons are too short for trinucleotide quadruplet analysis (Appendix A). With mononucleotide analysis, concatenated exon sequences longer than 6000 bp also show violation of CSPR (Figure 5F).

The results show higher frequencies of A than of T nucleotides in coding the DNA of all human chromosomes and in the coding DNA of all NBPF genes. This is a consequence of a proportionally higher presence of codons with two or three A nucleotides (stop codon 1/1 UAA, Asn 2/2 (AAC, AAU), Lys 2/2 (AAG, AAA), Glu 1/2 (GAA), Gln 1/2 (CAA), Ile 1/3 (AUA), Thr 1/4 (ACA), Arg 1/6 (AGA)), likely because of a greater need for encoded amino acids. For compensation, there is a higher relative percentage of T nucleotides in human chromosomes, which contain more than 98% of noncoding DNA, as well as in introns from NBPF genes. They preserve balance as an A-T complementary pair in the whole genome. Therefore, there is less than a 0.5% difference in the whole human chromosomes, which is hardly visible in Figure 3A, whereas less than 2% of differences in coding DNA are significantly higher (Figure 5F).

### 3.4. Are Some Rare Symbionts Exceptions to CSPR?

CSPR is present in all free-living bacteria, archaea, and eukaryotes. In prokaryotes, some of the following rare symbionts appear as possible exceptions: bacteria with extremely reduced genomes, with the smallest DNA among living species such as *C. tremblaya princeps* (138,927 bp) and *C. hodgkinia cicadicola* (143,795 bp), and bacteria with reduced genomes, such as *F. alocis* (1,931,012 bp) and *Pseudovibrio* (5,916,880 bp).

The symbiont *C. tremblaya princeps* has a coding part of 113,491 bp (which is 82% of the whole genome), whereas the noncoding part of 25,436 bp (which is 18% of the whole genome) is much smaller. *C. tremblaya princeps* tends to have approximate quadruplet symmetries (7/10 A+T-rich and 6/10 C+G-rich quadruplets) according to the CSPR principle [18,19]. As already pointed out, the method of determining relative frequencies with trinucleotide quadruplets is not adequate for the CSPR analysis of the noncoding part if it is too short. Therefore, in this case, we apply the method of dinucleotide quadruplets that sensitively tests CSPR in short sequences (Table 2). In this case, a difference between D and RC(D) for individual dinucleotides for noncoding DNA is in the interval of 0.06–0.52%. Furthermore, analysing mononucleotide frequencies in the same strand of DNA, we find that the noncoding part of the genome is in good accordance with CSPR. The deviation between A and T is only 0.34%, and between C and G, it is 0.49%. On the other hand, the coding part is not consistent with CSPR symmetry. This result is analogous to our results for human chromosomes and NBPF genes. We conclude that *C. tremblaya princeps* is not an exception with respect to free-living bacteria.

We also analyse the second smallest DNA genome in the symbiont *C. carsonella ruddi* (162,589 bp) [19].

Some nucleotide frequencies of *C. hodgkinia cicadicola, C. carsonela ruddi and F. alocis* in the coding part are larger than they are in the whole genome. In *Pseudovibrio_sp.FO-BEG1*, the difference between A-T and C-G complementary bases in the noncoding part is very high, but in the coding part, it is in good accordance with CSPR, which is unexpected (Appendix A). One could ask whether a possible mistake in determining the whole and coding DNA can contribute to disagreement between noncoding DNA and CSPR symmetry. Since the bacterial genome contains only one circular chromosome, in the analysis, it is likely that there is overlapping, so the coding part is doubly included. Due to uncertainties in available genomic data, there may be doubt regarding the reliability of whether CSPR is satisfied in this situation.

### 3.5. Possible Connections to the RNA World Hypothesis of DNA Creation

The RNA world hypothesis is a theoretical concept of evolution introduced in the 1960s [59,60,61,62], almost at the same time when Chargaff experimentally discovered CSPR symmetry. Interestingly, at that time, no attempt was made to consider possible connections between these two concepts. RNA has the difficulty of chemical fragility—it is unstable and catalytically limited. In a recent review, Bernhardt 2012 concluded that the RNA world hypothesis, “although far from perfect or complete, is the best we currently have to help understand the backstory to the contemporary biology” [63]. A crucial step is the transition from relatively unstable RNA to a more stable and robust two-stranded DNA molecule. This is reminiscent of Schrödinger´s intuitive idea proposing that the hereditary material must take the form of an “aperiodic crystal” [64], which could imply the presence of stability and symmetries in the structure of DNA for genetic information in living organisms. Besides the stabilizing effect of Watson–Crick base pairing, CSPR and the ensuing natural law of DNA creation and conservation with quadruplet symmetry, which act globally in a two-strand DNA system, could be a missing link in the mechanism of the RNA world hypothesis.

## 4. Conclusions

We can hypothesize that, after incorporating RNA segments into the structure of DNA, the natural law is activated, according to which the same mono- or oligonucleotide insertion must be inserted simultaneously into both strands of DNA in a bidirectional 5′3′↔3′5′ manner, but globally, i.e., regardless of the localization in the second strand, noncoding segments with more stable quadruplet symmetries are created. (Figure 6). In this way, identical complementary base pairs are inserted simultaneously into both strands as noncoding DNA, creating quadruplets with strict purine–pyrimidine symmetry, direct-complement symmetry on the principle of Watson–Crick pairing, and mirror symmetry. In conclusion, CSPR is the result of mirror symmetry based on the natural law of creation and conservation of DNA genomes [19,50,51].

Coding DNA, which is less than 2% of the human genome, represents small dispersed “islands” of exons embedded into the “ocean” of more than 98% noncoding DNA. Additionally, it is statistically more probable for changes to affect large introns than even smaller exons. An insertion can, in one step, occur in the exon, and in the next step, it can occur in the intron, reducing the CSPR effect on the gene without contributing to quadruplet symmetry and CSPR in the whole genome (Figure 5A–D). Namely, it follows that, regardless of possible insertions into genes, the primordial incorporated archaic RNA material, by itself, lacks organization of nucleotide pairs according to CSPR and quadruplet purine–pyrimidine symmetries, as seen in coding DNA. On the other hand, *ab ovo* insertions in accordance with the natural law for DNA creation and conservation persistently contribute to the build-up of noncoding sequences, creating quadruplet symmetries according to the CSPR principle, as empirically proved in almost all species. Finally, a small difference of about 0.5% in the relative frequencies of direct and reverse complement trinucleotides in human chromosomes is largely due to coding DNA.

Deviations from CSPR are the consequence of the insufficient length of investigated DNA sequences depending on the order of mono/oligonucleotides, which creates quadruplets. Namely, we observe that the minimal length *L_n_* required to test the applicability of CSPR to a given sequence for *n*-nucleotides/oligonucleotides is proportional to the number of different oligonucleotides of the order *n*.

However, we show that only coding DNA and genes do not satisfy quadruplet symmetry according to CSPR. Our results show higher frequencies of A than of T nucleotides in the coding DNA of all human chromosomes and in the coding DNA of all NBPF genes. As compensation, there is a higher relative percentage of T nucleotides in human chromosomes, with 98% of noncoding DNA and introns of all NBPF genes. They preserve balance as A-T complementary pairs and satisfy CSPR in the whole genome.

In conclusion, we can say that Watson–Crick pairing and the natural symmetry law of DNA creation and conservation with mirror symmetry result in CSPR and contribute to powerful evolutionary potential for the DNA molecule. In our analysis, we use the human genome as the entire thermodynamic system with a universal supersymmetry genetic code, which is common for all RNA and DNA species and unchangeable during evolution [51,52]. The genome’s stability is manifested in CSPR, which is present in all DNA species. CSPR, as the DNA symmetry of each genome, decreases disorder (entropy) and preserves the integrity of genomes. It is very important to see that, simultaneously, identical symmetry to CSPR also has the free energy of the trinucleotides/codons of each DNA genome and their supersymmetry genetic code [65,66,67].

The sophisticated structure of DNA based on symmetries may indicate that external interventions in DNA can contribute to symmetry violations with unpredictable consequences.

## Figures and Tables

**Figure 1 genes-13-01929-f001:**
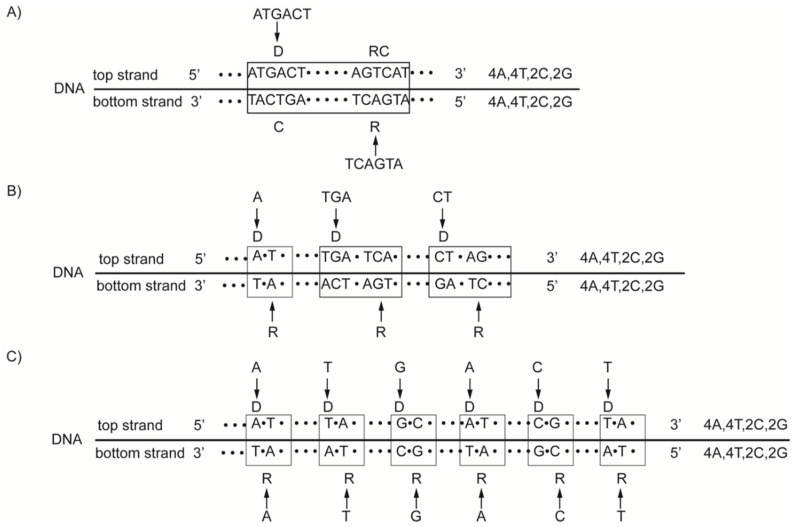
Examples for the natural symmetry law of DNA creation and conservation. According to this law, all mono/oligonucleotides which enter one strand of DNA must enter the second strand regardless of their localization. (**A**) Example with the entrance of a 6 nt oligonucleotide, ATGACT, into the top strand, reading its reverse oligonucleotide TCAGTA unidirectionally or reading the direct form of ATGACT bidirectionally (5′3′↔3′5′), entering the bottom strand. Thus, mirror symmetry between both strands is created. Due to Watson–Crick pairing and mirror symmetry, the quadruplet structures are formed, automatically fulfilling CSPR: *f*D = *f*RC. (**B**) The same nucleotides may also enter as mononucleotide (A), trinucleotide (TGA) and dinucleotide (CT). The farther process and result are identical, as in A. (**C**) The same 6 nucleotides can enter the top strand and the bottom strand individually as mononucleotides. Binding with a complementary pair, the quadruplet structures are created, and the final CSPR result is identical, as in cases A and B. It can be seen how the bidirectional DNA structure is formed: AT(D) in the top strand and TA(R) in the bottom strand, or GC(D) in the top strand and CG(R) in the bottom strand.

**Figure 2 genes-13-01929-f002:**
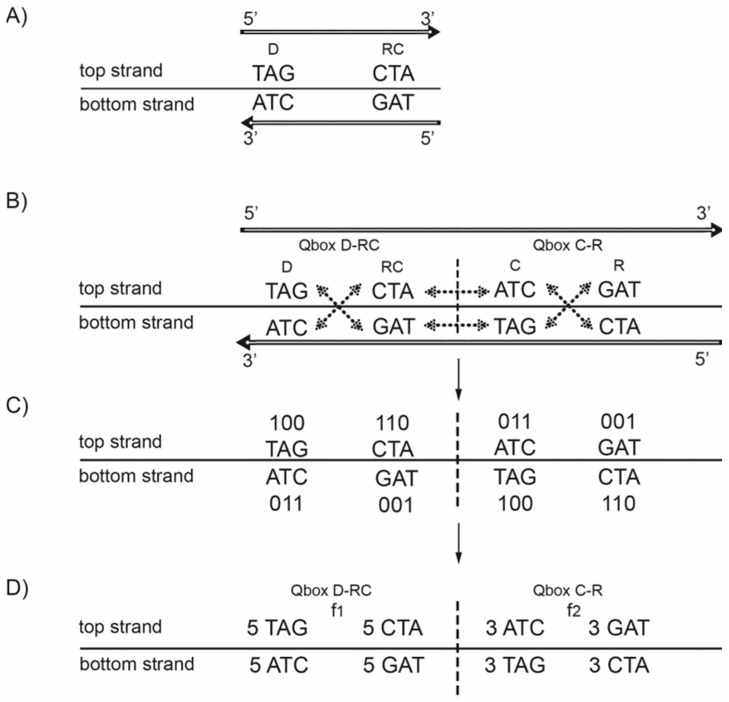
The difference between strand symmetry and quadruplet symmetry for triplets. (**A**) Strand symmetry includes in the same strand of only the direct (D) and reverse complement (RC) of a triplet. Reading bidirectionally, as pointed by the direction of the arrow, the same trinucleotides appear in both strands. Therefore, only one strand is considered in the determination of strand symmetry (the top strand). Thus, the DNA is reduced to a binary system. However, in this way, symmetries among trinucleotides are not evident. (**B**) Quadruplet symmetry includes the whole quadruplet of trinucleotides: direct (D), reverse complement (RC) as well as the complement (C) and reverse (R) in both strands of DNA. The quadruplet boxes QboxD-R and QboxC-R are created with mirror symmetries between the strands and between the boxes. Thus, each quadruplet consists of structural symmetries, creating an esthetical form of “butterfly” mirror symmetry, and the DNA is reduced to a quartic system. (**C**) The same quadruplet mirror symmetries are present in the purine–pyrimidine relationship: 0 is assigned to purines (A, G), and 1 is assigned to pyrimidines (T, C). (**D**) All members of the same box have the same frequencies (*f*D = *f*RC, respectively, *f*C = *f*R), but frequencies between the boxes mutually differ. For quadruplets with symmetric trinucleotides, such as AGA or CTC, there is no difference in frequencies between boxes. However, frequencies in both strands of DNA for each individual quadruplet are identical regardless of whether the trinucleotides are symmetric or asymmetric (*f*D = *f*RC = *f*C = *f*R), such as in our example: (D) 5 TAG (top strand) + 3 TAG (bottom strand) = 8 TAG; (RC) 5 CTA (top strand) + 3 CTA (bottom strand) = 8 CTA; (C) 5 ATC (bottom strand) + 3 ATC (top strand) = 8 ATC; and (R) 5 GAT (bottom strand) + 3 GAT (top strand) = 8 GAT.

**Figure 3 genes-13-01929-f003:**
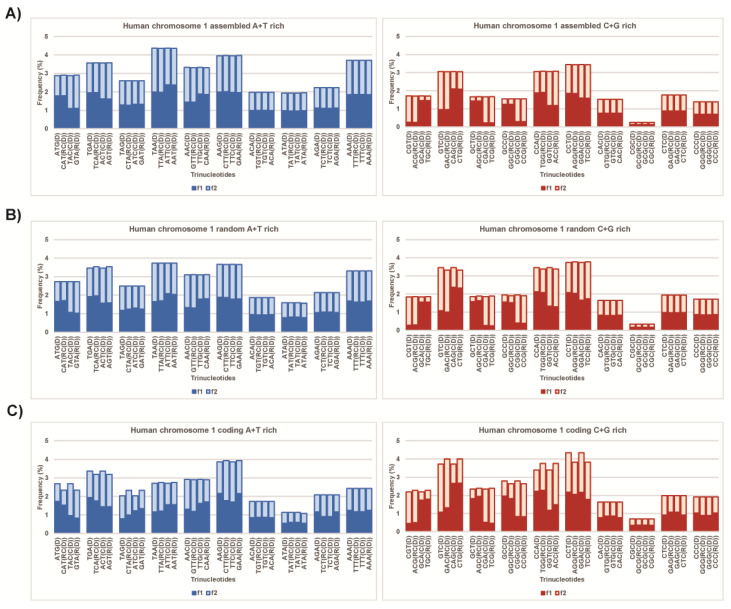
A-T-rich and C-G-rich trinucleotide quadruplet matrices of human chromosome 1. (**A**) In each quadruplet, the frequency of all four members (D, RC, C and R) in both strands is practically identical (for numerical values, as reflected in the plateau on the upper edge of each quadruplet. The plateau shows that the investigated sequence (chromosome, genome) is in accordance with CSPR. (**B**) Quadruplet matrices in random sequences of 200,000 bp. A concatemer of 1000 randomly selected 200 bp subsequence from human chromosome 1 shows very small deviations in some quadruplets (smaller than 1%), which is reflected by a slightly indented plateau, showing approximate agreement with CSPR. (**C**) Quadruplet matrices for concatenated coding sequences longer than 17,000,000 bp of the same chromosome show significant deviations in frequencies and no agreement with CSPR, especially for asymmetric trinucleotides. Instead of a plateau, the frequency for D and C differs from the frequency for R and RC, only because of Watson–Crick pairing.

**Figure 4 genes-13-01929-f004:**
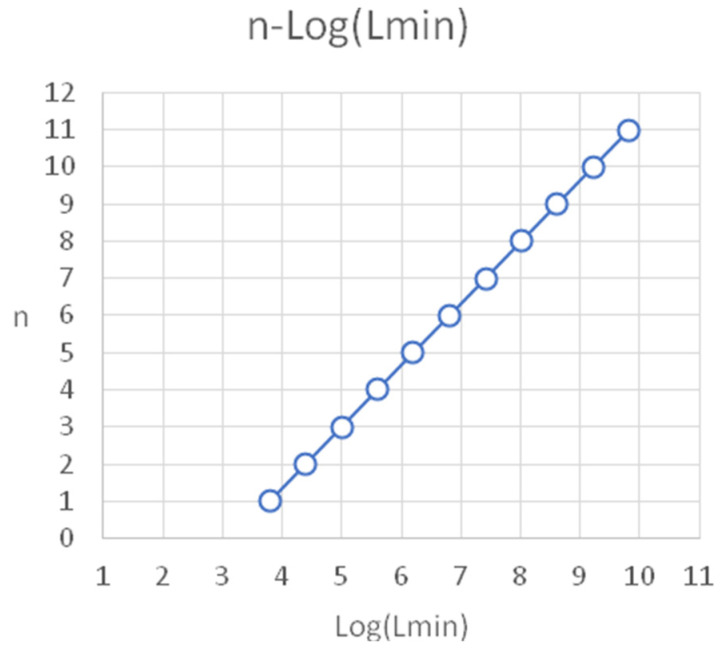
Diagrammatic presentation of the logarithmic relation between oligonucleotides of order *n* vs. the minimal length *L*_min_ of a human genome sequence (columns 1 and 4 from Table 2). The graph is a straight line, parallel to the diagonal. Changes in empirical genomic length were used to gauge the results of translating a straight line parallel to itself.

**Figure 5 genes-13-01929-f005:**
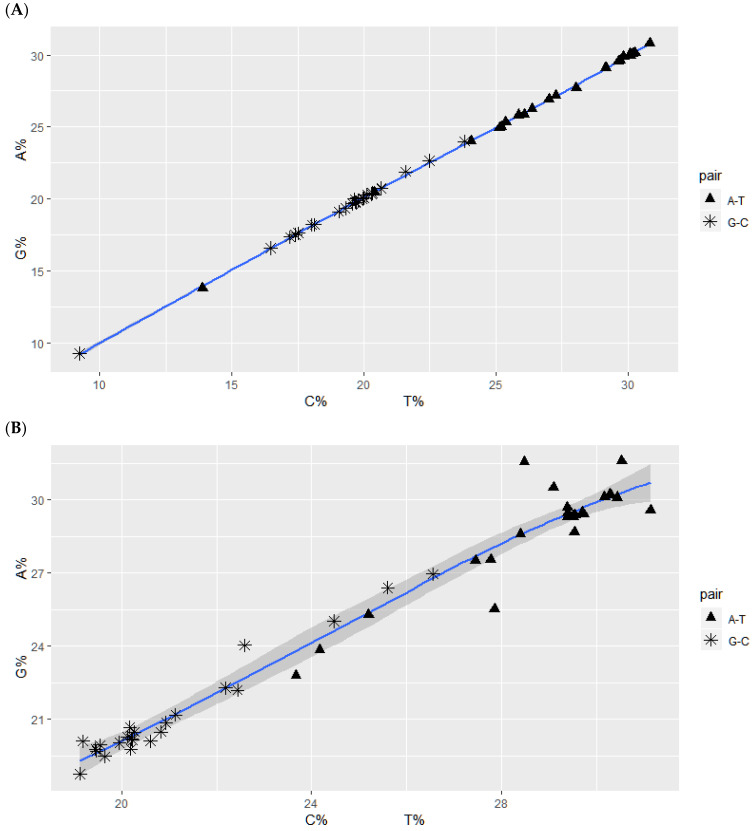
Relative frequencies of nucleotides in complementary base pairs of human chromosomes. Vertical axis: purines A and G; Horizontal axis: complementary pyrimidines T and C. For the case of exactly fulfilled CSPR, the base pairs A-T and G-C lie on a diagonal straight line, *f*A ≈ *f*T, *f*G ≈ *f*C, in the purine–pyrimidine diagram. Deviations from CSPR cause deviations from diagonality. Numerical values are given in Appendix A. (**A**) Relative frequencies of complementary nucleotide pairs in human chromosomes. The purine–pyrimidine frequency graph is very close to a typical diagonal straight line of exact CSPR. It is used for comparison with the graphs in Figure 5B–F; Appendix A. Confidence interval (0.9996, 0.9999) and Pearson’s product-moment correlation cor value: 0.9998. (**B**) Comparison of relative frequencies of complementary nucleotide pairs in a random sequence of 200,000 bp. A total of 1000 randomly selected 200 bp subsequences from each human chromosome are concatenated. The purine–pyrimidine frequency graph is rather close to a diagonal straight line of exact CSPR. Confidence interval (0.9721, 0.9912) and Pearson’s product-moment correlation cor value: 0.9843. (**C**) Comparison of relative frequencies of complementary nucleotide pairs from the coding DNA of each chromosome. Relative frequencies of nucleotides in complementary nucleotides differ rather significantly (*f*A ≠ *f*T, *f*G ≠ *f*C), as seen by the wave-like deviation of the purine–pyrimidine frequency graph from the CSPR diagonal straight line. The relative frequencies of A nucleotides are systematically higher than those of T nucleotides in all chromosomes. Confidence interval (0.8747, 0.9592) and Pearson’s product-moment correlation cor value: 0.9281. (**D**) Comparison of relative frequencies of complementary nucleotide pairs from all NBPF genes with introns included in human chromosome 1. Confidence interval (0.6211, 0.8732) and Pearson’s product-moment correlation cor value: 0.7766. Each gene does not satisfy CSPR. (**E**) Comparison of relative frequencies of complementary nucleotide pairs from introns from NBPF genes in human chromosome 1. Values of complementary pairs differ sizably, and there is substantial deviation from CSPR. Introns even longer than 100,000 bp do not satisfy CSPR. Confidence interval (0.5532, 0.8485) and Pearson’s product-moment correlation cor value: 0.7338. Relative frequencies of T nucleotides are systematically higher than those of A nucleotides in all NBPF genes. (**F**) Comparison of relative frequencies of complementary nucleotide pairs from exons from NBPF genes in human chromosome 1. Values of complementary pairs largely differ, and there is a strong deviation from CSPR. Exons of 6000 bp or more do not satisfy CSPR. Confidence interval (−0.6375, −0.1406) and Pearson’s product-moment correlation cor value: −0.4200. The slope of the graph in (**F**) is decreasing (negative), which is the opposite compared to graphs (**A**–**E**) because of significantly higher values of A with respect to T nucleotides.

**Figure 6 genes-13-01929-f006:**
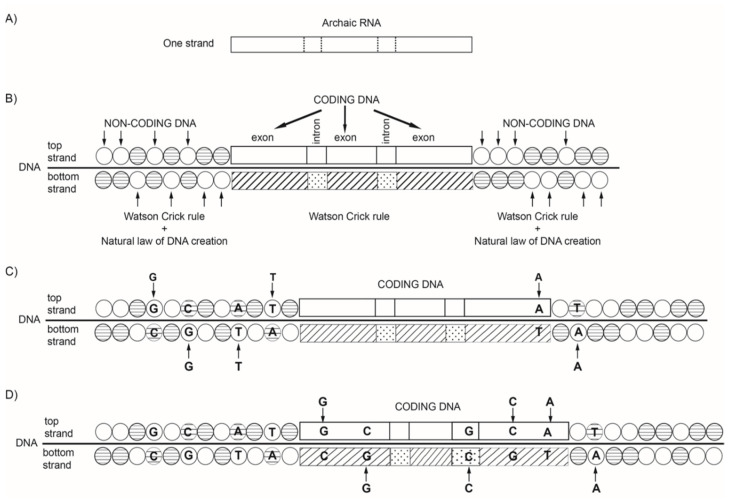
Novel hypothesis of DNA creation. (**A**) Single strand archaic RNA, being short, fragile and without symmetries and with the absence of CSPR, had limited evolutionary potential. (**B**) Double-stranded DNA originated by binding archaic RNA with its complementary nucleotides, creating pairs A→T, C→G according to the Watson Crick rule, giving rise to the double helix of coding DNA. It was initially also symmetry-free, with the absence of CSPR. Simultaneously, besides the Watson–Crick rule, there appeared a new insertion of nucleotides according to the natural symmetry law of DNA creation and conservation; the same nucleotides entered both strands, regardless of localization. This led to the creation of noncoding DNA with quadruplet mirror symmetries. (**C**) As a result of the new principle of creation based on the natural symmetry law of DNA creation and conservation, noncoding DNA grows much faster, and thus, the new insertions are induced. If an additional insertion enters a segment on one strand of coding DNA, it contributes to its enlargement but without creating new symmetries. On the other hand, the same insertion into the long noncoding part of the DNA does not essentially violate its symmetries. (**D**) Insertion into one strand of an exon, and according to the natural symmetry law of DNA creation and conservation, the accompanying insertion into the opposite strand but in the position of an intron does not create symmetries and CSPR for coding DNA. If the same insertions enter both strands of the exon, they do not lead to symmetries and CSPR because they do not compensate symmetry-free archaic DNA built into the DNA molecule. In summary, insertions, as rare events, cannot violate symmetries in large noncoding DNA because of the natural symmetry law of DNA creation and conservation, and insertions (mutations) in much smaller coding DNA cannot create symmetries because of built-in symmetry-free archaic RNA. However, viewing the whole DNA molecule, regardless of whether an insertion enters the coding or noncoding part, due to the natural symmetry law of DNA creation and conservation, the same insertions always enter both strands, and thus, its growth fully supports CSPR. In this way, quadruplets, as basic building elements with mirror symmetries, gave the DNA molecule strong evolutionary potential.

**Table 1 genes-13-01929-t001:** Our novel quadruplet classification of all 64 possible trinucleotides. A quadruplet is a basic structure of a genome. Each quadruplet is unique and consists of nucleotides denoted as direct **(D)**, its reverse complement (RC), the complement (C) and reverse (R). The number 0 is assigned to purine, and 1 is assigned to pyrimidine. Trinucleotides have 10 A+T-rich and 10 C+G-rich quadruplets. The C+G-rich trinucleotides (group II) correspond to the purine–purine and pyrimidine–pyrimidine transformation of A+T-rich trinucleotides (group I). Within quadruplets, each member can be chosen as direct, and the other three can be arranged according to Watson–Crick pairing. Therefore, each quadruplet contains the same number of A and T as well as C and G bases, automatically satisfying CSPR. Quadruplets are ordered in rows, emphasizing purine–pyrimidine symmetry.

A+T-Rich Group (I)				C+G-Rich Group (II)			
D	RC(D)	C(D)	R(D)	D	RC(D)	C(D)	R(D)
**TGA** 100	TCA 110	ACT 011	AGT 001	**CAG** 100	CTG 110	GTC 011	GAC 001
**TAG** 100	CTA 110	ATC 011	GAT 001	**CGA** 100	TCG 110	GCT 011	AGC 001
**TAA** 100	TTA 110	ATT 011	AAT 001	**CGG** 100	CCG 110	GCC 011	GGC 011
**CAA** 100	TTG 110	GTT 011	AAC 001	**TGG** 100	CCA 110	ACC 011	GGT 001
**ATG** 010	CAT 101	TAC 101	GTA 010	**GCA** 010	TGC 101	CGT 101	ACG 010
**ATA** 010	TAT 101	TAT 101	ATA 010	**GCG** 010	CGC 101	CGC 101	GCG 010
**ACA** 010	TGT 101	TGT 101	ACA 010	**GTG** 010	CAC 101	CAC 101	GTG 010
**AGA** 000	TCT 111	TCT 111	AGA 000	**GAG** 000	CTC 111	CTC 111	GAG 000
**AAG** 000	CTT 111	TTC 111	GAA 000	**GGA** 000	TCC 111	CCT 111	AGG 000
**AAA** 000	TTT 111	TTT 111	AAA 000	**GGG** 000	CCC 111	CCC 111	GGG 000

**Table 2 genes-13-01929-t002:** Estimates of minimal length *L*_min_ of a given DNA sequence, for which CSPR (strand symmetry) still approximately persists using an *n*-nucleotide oligonucleotide. As the reference value that we used for trinucleotides (*n* = 3), we empirically estimated the minimal length of a genomic sequence (~100,000 bp). It follows that, for the estimate in this minimal sequence, each trinucleotide must be present ~1500 times. Using this estimate as a gauge, we determined estimates for minimal lengths of oligonucleotides of other orders.

Length of Oligonucleotide *n*	No. of Different Oligonucleotides of Length *n*	Estimated *L*_min_ bp	Log *L*_min_
**1**	4	6250	3.80
**2**	16	25,000	4.40
**3**	64	100,000 *	5.00
**4**	256	400,000	5.60 n
**5**	1024	1,600,000	6.20
**6**	4096	6,400,000	6.81
**7**	13,384	25,600,000	7.41 g
**8**	65,536	102,400,000	8.01
**9**	262,144	409,600,000	8.61
**10**	1,048,576	1,638,400,000	9.21
**11**	4,194,304	6,553,600,000	9.82

* Empirical estimate for trinucleotides, used to gauge sequence length for oligonucleotides of other orders; 4*^n^*—number of different oligonucleotides built from *n* nucleotides; estimated value *L*_min_ = 1500 … 4*^n^*^−3^—minimal length of genomic sequence for which CSPR (strand symmetry) persists.

## Data Availability

Our data sets are described in materials and methods.

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
