# Peer review of "An Explanation of Exceptions from Chargaff’s Second Parity Rule/Strand Symmetry of DNA Molecules"

_genes, 2022, doi:10.3390/genes13111929_

Round 1

Reviewer 1 Report

The author of the manuscript explored exceptions from Chargaff's second parity rule and strand symmetry of DNA molecules using publically available genomes of multiple organisms with a computational method. 

Although the method and reasoning behind the analysis are sound, I have two major concerns about the paper.

1. Regarding the human genome used in this paper, hg38,p7, is now outdated. The result presented showed that the coding part of the human genome follows Chargaff's rule, while the genome shows some exceptions. This could be due to the vias for the coding sequences during the sequencing and the difficulty of sequencing the highly repeated sequences. The exception could be due to the sequencing platform. I suggest the authors either incorporate that possibility or run an additional analysis with the new human genome (T2T-CHM13v2.0).

2. The second concern is with the method. Although it is proprietary, Python or other analysis scripts should be available for others to download and all the parameters should be clearly defined. 

Author Response

Reply to reviewer 1 and changes in manuscript

Ad 1) Computations investigating preservation/violation of Chargaff՚s second parity rule (CSPR) are investigated the hg38 human genome assembly. 

INSERT IN MANUSCRIPT:

Additional insert for revised manuscript:         

We have shown that CSPR symmetry analysis of triplets in hg38 human genomic sequences preserves the CSPR as a global pattern in noncoding part of DNA and violates CSPR in the coding part.  We have used the hg38 assembly because when we started with this study, the available assembly sequence was incomplete. Only recently, complete reference assembly T2T-CHM13 opens new opportunities to investigate the role of genome organization and regulation (Miga, 2020; Nurk et al., 2022).  We argue that extension of analysis to T2Tassembly will not significantly influence our results obtained by using the hg38 genomic assembly. For coding parts of DNA this is obvious, since genes are mostly nearly gap-free and well sequenced already in hg38, giving similar results as T2T assembly. For example, we have checked this for NBPF genes with relatively long NBPF exons in human chromosome 1. In sizeable sequenced noncoding segments, no CSPR violations are found, both in continuous arrays and in concatenated widely scattered randomized segments. In many of such cases these structures are characterized by repeat pattern and their deviations, of similar type in hg38 sequenced sections and those which are sequenced only by T2T. Such comparisons have been considered, for example, for human chromosomes 21 and Y.  (Insertion line from 196 to 215).

Ad 2) Computational method CLT__Find used in calculations is available at server http://genom.hazu.hr/tools.html.

Reply to reviewer 1 and 2 and changes in manuscript

We used human genome as thermodynamic system with universal Supersymmetry genetic code which is common for all RNA and DNA species and unchangeable during evolution (Rosandić and Paar 2022a, Rosandić 2022). The genome stability is manifested in CSPR which is present in all DNA species. CSPR symmetry of each genome decreases disorder (entropy) and preserves integrity of genomes. It is important to see that simultaneously identical symmetry as CSPR have also free energy of trinucleotides/codons of each DNA genome and Supersymmetry genetic code (Breslauer et al. 1986; Klump et al. 2020; Rosandić and Paar 2022B). (Insertion line from 552 to 559).

Additional references:

Breslauer KJ, Frank R, Blocker H, Marky LA (1986) Predicting DNA duplex stability from the base sequence. Proc Natl Acad Sci USA, doi.org/10.1073/pnas.83.113746.

Klump H H, Volker J, Breslauer KJ (2020) Energy mapping of the genetic code3 and genomic domains: implications for code evolution and molecular Darwinism. Cambridge University Press, doiu.org/10.101/S0033583520000098.

Miga K H (2020) Centromere studies in the era of “telomere-to-telomere” genomics. Exp Cell Res 394, 112127.

Nurk S et.al. (2022) The complete sequence of human genome. Science 376, 44-53.

Rosandić M and Paar V (2022b) Supersymmetry genetic code table and quadruplet symmetries of DNA molecule are unchangeable and synchronized with the codon free energy mapping during evolution. submitted to J Theor Biol

Rosandić M (2022) Ryugu amino acid samples support genetic code supersymmetry. Science eLetters (1).

https://doi.org/10.1126/science.abn7850

Reviewer 2 Report

The manuscript, entitled EXPLANATION OF EXCEPTIONS FROM CHARGAFFS SECOND PARITY RULE/STRAND SYMMETRY OF DNA MOLECULE (genes-1950223), evaluates Chargaff's second parity rule (CSPR) in five genomes covering Carsonella ruddii, Tremblaya princeps, Filifactor alocis, Pseudovibrio_sp.FO-BEG1 and human genomes and human NBPF family genes. The authors classified oligonucleotide steps into quadruplets based on their orientation in the DNA strands, namely direct:D, reverse complement:RC(D), complement:C(D), and reverse R(D) units. They claimed that the human genome (excluding coding DNA and genes) adheres to quadruplet symmetry according to CSPR. The authors propose a missing link in the mechanism of the RNA world hypothesis that, together with Watson-Crick base pairing, CSPR, and conservation with quadruple symmetry, ensues the natural law of DNA generation from archaic RNA. The research presented in the article appears to be very interesting and may provide additional insights into the evolution of genomes. However, the work lacks an adequate presentation of results, discussion or conclusions and is premature to publish in its current form. Authors should carefully address the following concerns to improve their manuscript.

Major concerns:

1.    The simple classification of genomic DNA into coding and non-coding does not support the author's hypothesis. Non-coding DNA may be functional or non-functional.  Functional non-coding DNA includes DNA  that transcribes into f non-coding RNA molecules namely transfer RNA, microRNA, piRNA, ribosomal RNA, and regulatory RNAs or regulatory sequences that control gene expression namely promoters, origins of DNA replication, centromeres, scaffold attachment regions, and telomeres. However non--functional regions include introns, pseudogenes, intergenic DNA, and transposons. The manuscript should incorporate additional results and discussion in the mentioned context.

2.    Genomes are considered as continuous thermodynamic landscapes, multiple codes exist in genomes, namely genetic code, cis-regulatory code and nucleosome positioning code. Authors should link their work accordingly.

3.    The authors chose only five genomes and the criterion for selecting very limited datasets is not justified in the manuscript.  Otherwise. The analysis seems to be incomplete. The analysis appears to be incomplete. Authors should discuss what new knowledge this research provides in comparison to their previous work or similar published research by others.

4.    The novel quadruplet classification of all 64 possible trinucleotides has not been unambiguously presented. The rationale for higher order oligonucleotides is not clear from Table 2.

5.    Authors should incorporate comparative analysis of shuffled or randomized genomic sequences with real genomic sequences to provide solid support to their conclusions.

6.    Authors should reconsider the headings for each section in the results and discussion.

7.    The manuscript is poorly written except for the introduction. There are several typographical, and grammatical errors and sweeping statements throughout the manuscripts.   

Author Response

Reply to reviewer 2 and changes in manuscript

Ad 1)                                                                                                                        For Chargaff՚s second parity rule (CSPR) symmetry is not important whether a noncoding DNA region is functional or nonfunctional. In both cases the trinucleotide symmetries in CSPR are identical to symmetries of free energies of trinucleotides/codons (Rosandić and Paar, 2022B), and have a role in decreasing disorder (entropy) in the whole genome.

Ad 2)                                                                                                                      In our analysis we use the physicochemical Supersymmetry genetic code table, which is common for all RNA and DNA species and unchangeable during evolution (Rosandić and Paar, 2022 A, Rosandić 2022)).  

Ad 3)                                                                                                                                                                                      Large exceptions from CSPR are rare. We have considered previously available genomes of five symbionts as such cases. However, in three of these genomes there was a problem with sequencing, with the number of nucleotides in coding part larger than their reported number for whole genome, or large deviation from CSPR in small noncoding part. 

Ad 4, 5)                                                                                                                                                                                      Among oligonucleotides, mostly trinucleotides were studied here in relation to CSPR and quadruplets. It was empirically shown for trinucleotides that a minimal length of the genomic segment required to determine the CSPR symmetry should be about 100 kb. This means that in a sequence of minimal length 100 kb each trinucleotide is presented about 1500 times (100,000 / 64 = 1562). Using this estimate as a gauge we determined minimal lengths of DNA sequences of the order 2, 3, 4, … to 11 (Table 2). The main new knowledge from this work in comparison with previous investigations is the discovery the Chargaff՚s second parity rule is violated in coding (exsons and introns) sequences. This is demonstrated especially for the whole human genome (hg38 assembly), for NBPF genes which are characterized by exons of sizable length, and for concatenated randomized 200-kb noncoding subsequences from all human chromosomes (Fig. 4, Supplementary Table 2 and 7b). As illustrations of possible exceptions to CSPR for single mononucleotides we consider and discuss five symbionts with possible degree of symmetry breaking, especially Candidatus tremblaya princeps (138,795 bp) and Pseudovibrio sp FO-BEG1 (1,346,359 bp).    

Ad 6. All headings of subsections in the section “Results and Discussion” are bolded” to make the manuscript more transparent.

Ad 7) Several typographical and grammatical errors and misprints throughout the manuscript are corrected.

Reply to reviewer 1 and 2 and changes in manuscript

We used human genome as thermodynamic system with universal Supersymmetry genetic code which is common for all RNA and DNA species and unchangeable during evolution (Rosandić and Paar 2022a, Rosandić 2022). The genome stability is manifested in CSPR which is present in all DNA species. CSPR symmetry of each genome decreases disorder (entropy) and preserves integrity of genomes. It is important to see that simultaneously identical symmetry as CSPR have also free energy of trinucleotides/codons of each DNA genome and Supersymmetry genetic code (Breslauer et al. 1986; Klump et al. 2020; Rosandić and Paar 2022B). (Insertion line from 552 to 559).

Additional references:

Breslauer KJ, Frank R, Blocker H, Marky LA (1986) Predicting DNA duplex stability from the base sequence. Proc Natl Acad Sci USA, doi.org/10.1073/pnas.83.113746.

Klump H H, Volker J, Breslauer KJ (2020) Energy mapping of the genetic code3 and genomic domains: implications for code evolution and molecular Darwinism. Cambridge University Press, doiu.org/10.101/S0033583520000098.

Miga K H (2020) Centromere studies in the era of “telomere-to-telomere” genomics. Exp Cell Res 394, 112127.

Nurk S et.al. (2022) The complete sequence of human genome. Science 376, 44-53.

Rosandić M and Paar V (2022b) Supersymmetry genetic code table and quadruplet symmetries of DNA molecule are unchangeable and synchronized with the codon free energy mapping during evolution. submitted to J Theor Biol

Rosandić M (2022) Ryugu amino acid samples support genetic code supersymmetry. Science eLetters (1).

https://doi.org/10.1126/science.abn7850

Round 2

Reviewer 2 Report

The authors have satisfactorily addressed my concerns. However, I suggest careful checking for typographical, punctuation, and grammatical errors to improve your manuscript (use consistent abbreviations in table and column headings).

Author Response

Dear reviewer, we correct tipfelers and added a word in lin 488 - very high.